# Challenges of Measuring Soluble Mn(III) Species in Natural Samples

**DOI:** 10.3390/molecules27051661

**Published:** 2022-03-03

**Authors:** Bohee Kim, Usha Farey Lingappa, John Magyar, Danielle Monteverde, Joan Selverstone Valentine, Jaeheung Cho, Woodward Fischer

**Affiliations:** 1Department of Emerging Materials Science, Daegu Gyeongbuk Institute of Science and Technology, Daegu 42988, Korea; bkim97@jh.edu; 2Division of Geological and Planetary Sciences, California Institute of Technology, Pasadena, CA 91125, USA; usha@caltech.edu (U.F.L.); jmagyar@caltech.edu (J.M.); dmonteverde@caltech.edu (D.M.); jsv@chem.ucla.edu (J.S.V.); 3Department of Chemistry and Biochemistry, University of California Los Angeles, Los Angeles, CA 90095, USA; 4Department of Chemistry, Ulsan National Institute of Science and Technology, Ulsan 44919, Korea

**Keywords:** manganese, leucoberbelin-blue method, porphyrin method, desferrioxamine B

## Abstract

Soluble Mn(III)–L complexes appear to constitute a substantial portion of manganese (Mn) in many environments and serve as critical high-potential species for biogeochemical processes. However, the inherent reactivity and lability of these complexes—the same chemical characteristics that make them uniquely important in biogeochemistry—also make them incredibly difficult to measure. Here we present experimental results demonstrating the limits of common analytical methods used to quantify these complexes. The leucoberbelin-blue method is extremely useful for detecting many high-valent Mn species, but it is incompatible with the subset of Mn(III) complexes that rapidly decompose under low-pH conditions—a methodological requirement for the assay. The Cd-porphyrin method works well for measuring Mn(II) species, but it does not work for measuring Mn(III) species, because additional chemistry occurs that is inconsistent with the proposed reaction mechanism. In both cases, the behavior of Mn(III) species in these methods ultimately stems from inter- and intramolecular redox chemistry that curtails the use of these approaches as a reflection of ligand-binding strength. With growing appreciation for the importance of high-valent Mn species and their cycling in the environment, these results underscore the need for additional method development to enable quantifying such species rapidly and accurately in nature.

## 1. Introduction

The chemistry of manganese associated with biomass, Earth surface environments, and geological materials is characterized by redox conversions among three common oxidation states [1]. The reduced form, Mn(II), is highly soluble in water, but, when oxidized, it tends to form insoluble Mn(III) and (IV) oxide minerals. In the absence of stabilizing ligand coordination, any Mn(III) in solution will rapidly disproportionate to form Mn(II) and Mn(IV) oxide [2]. However, with the coordination of appropriate ligands, this disproportionation reaction can be slowed, allowing Mn(III) to persist in solution long enough to participate in other reactions. Recent years have seen a growing body of work demonstrating that such Mn(III) species constitute a significant proportion of the soluble manganese pool in natural environments [3,4,5,6] and uncovering a broad suite of implications for this element’s role in aquatic chemistry and biogeochemistry [7,8,9,10,11,12,13].

Mn(III) complexes can be highly reactive, making them important and dynamic players in a myriad of biogeochemical processes. As uniquely high-potential single-electron oxidants, such species contribute to the breakdown of otherwise recalcitrant molecules. For example, Mn(III)–oxalate generated by fungal Mn peroxidases controls lignin decomposition in the leaf litter of forest ecosystems—a critical process in carbon cycling [7,14]. Similarly, Mn(III)–humic acid complexes are likely responsible for the degradation of anthropogenic pollutants, including estrogens in natural waters [10]. These processes are based on dynamic Mn redox cycling in which the Mn(III) complexes themselves are reactive and transient. They may exist in substantial steady-state concentrations, but only through constant fluxes of formation.

It is important to recognize that Mn(III) complexes behave very differently from similar ligand complexes with ferric iron (Fe(III)). Due to the relative energies of their respective di-, tri-, and tetravalent redox states, Mn(III) is reactive and unstable in ways that Fe(III) is not [13]. Mn(III) complexes are susceptible to both inter- and intramolecular electron transfer and ligand decomposition, and, therefore, they tend to be very short-lived chemical species unless stabilized by special redox inert ligands. The different modes by which Mn(III) complexes decompose were nicely illustrated in a study by Klewicki and Morgan that examined the behavior of the Mn(III) complexes of pyrophosphate (PP), ethylenediaminetetraacetic acid (EDTA), and citrate [15]. The Mn(III) complex of PP—a ligand that does not undergo intramolecular redox reactions [6]—displayed the most stability, decomposing slowly by PP hydrolysis and Mn(III) disproportionation on a timescale of months. By contrast, the Mn(III) complexes of EDTA and citrate decomposed readily by internal electron transfer to generate Mn(II) and products of ligand oxidation, on timescales ranging from minutes to days [15,16]. The Mn(III) complex of desferrioxamine B (DFOB) was shown by Duckworth and Sposito to decompose similarly by internal electron transfer to give Mn(II) and an oxidized derivative of DFOB at rates that rapidly increased with decreasing pH (half-life of ~24 h at pH 6, but less than one hour at pH 5.7) [17]. These studies demonstrated that, in addition to bimolecular disproportionation, intramolecular electron-transfer processes are an important aspect of the reactivity of Mn(III) species, particularly when complexed to organic ligands. When conceptualizing fluxes and roles for such complexes in the environment, this chemistry should be taken into account.

Due to this innate reactivity and lability, designing methods for measuring soluble Mn(III) species in environmental or experimental samples requires consideration of these factors for each individual type of sample tested [18]. Transient species can be lost in the time required to transport samples collected in the field back to the laboratory for analysis. Therefore, spectrophotometric methods that are adaptable to rapid field-based measurements have considerable appeal. Two such methods have been particularly important: one using the dye leucoberbelin blue (LBB), and the other using a Cd(II)–porphyrin, α,β,γ,δ-tetrakis(4-carboxyphenyl)porphyrin (TCPP), complex (Figure 1). It is, therefore, necessary to understand the degree to which both of these methods have material limitations in their application to quantifying Mn(III) complexes in environmental samples.

The LBB method (Figure 1A,C) has been used to detect Mn of any redox state higher than (II), i.e., Mn(III) to Mn(VII) [19,20,21,22]. With this approach, LBB is oxidized by high-valent Mn to form a product with a strong characteristic absorbance at 624 nm. Since the magnitude of this response is stoichiometric with electrons transferred, LBB can be considered a redox titration method. It cannot be used to determine absolute concentrations of Mn species of unknown redox state, but it can provide a measurement of average Mn redox state in a sample when combined with other measurements for total Mn concentration [23]. In natural environments, the LBB-reactive Mn pool largely comprises particulate Mn oxides and soluble Mn(III)–L complexes. Therefore, in samples that are filtered to separate a soluble fraction from a particulate fraction, LBB reactivity has been interpreted to reflect soluble Mn(III) complexes [24]. (However, it is important to note that filtration leads to an operational definition of solubility that does not necessarily exclude colloidal or nanoparticulate Mn phases, which may also be biologically and geochemically important, reactive Mn species [25,26,27].)

The TCPP method (Figure 1B,D) has been employed to determine concentrations of both Mn(II) and Mn(III). This method was originally designed to detect Mn(II) [28], and it was more recently adapted to measure Mn(II) and Mn(III) simultaneously in the same samples [29]. In the case of Mn(II), Mn(II) reacts with Cd(II)–TCPP by a metal-substitution reaction to yield Mn(II)–TCPP. Once complexed by the TCPP, the Mn(II) is rapidly oxidized by ambient dioxygen (O_2_) to generate Mn(III)–TCPP, which has a strong characteristic absorbance at 468 nm. In the case of Mn(III), Mn(III) has been proposed to react with the Cd(II)–TCPP by a ligand exchange reaction, also to yield Mn(III)–TCPP. Both of these reactions were considered as (pseudo)first-order reactions, but with different rate constants—rapid for Mn(II) and slower for Mn(III), i.e., k_Mn(II)_ > k_Mn(III)_:(1)$$\frac{d{\left[\mathrm{Mn}\left(\mathrm{III}\right)\mathrm{TCPP}\right]}_{\mathrm{from}\;\mathrm{Mn}\left(\mathrm{II}\right)}}{\mathrm{dt}}=-\frac{d\left[\mathrm{Mn}\left(\mathrm{II}\right)\right]}{\mathrm{dt}}=k_{\mathrm{Mn}\left(\mathrm{II}\right)}\left[\mathrm{Mn}\left(\mathrm{II}\right)\right]$$
(2)$$\frac{d{\left[\mathrm{Mn}\left(\mathrm{III}\right)\mathrm{TCPP}\right]}_{\mathrm{from}\;\mathrm{Mn}\left(\mathrm{III}\right)}}{\mathrm{dt}}=-\frac{d\left[\mathrm{Mn}\left(\mathrm{III}\right)\right]}{\mathrm{dt}}=k_{\mathrm{Mn}\left(\mathrm{III}\right)}\left[\mathrm{Mn}\left(\mathrm{III}\right)\right]$$
Integrating on time yielded:(3)$${\left[\mathrm{Mn}\left(\mathrm{III}\right)\mathrm{TCPP}\right]}_{\mathrm{from}\;\mathrm{Mn}\left(\mathrm{II}\right)}={\left[\mathrm{Mn}\left(\mathrm{II}\right)\right]}_{\mathrm{initial}}\left(1-e^{-k_{\mathrm{Mn}\left(\mathrm{II}\right)}t}\right)$$
(4)$${\left[\mathrm{Mn}\left(\mathrm{III}\right)\mathrm{TCPP}\right]}_{\mathrm{from}\;\mathrm{Mn}\left(\mathrm{III}\right)}={\left[\mathrm{Mn}\left(\mathrm{III}\right)\right]}_{\mathrm{initial}}\left(1-e^{-k_{\mathrm{Mn}\left(\mathrm{III}\right)}t}\right)$$
Which, when solved and summed for the total ingrowth of Mn(III)–TCPP over time, gave the following:(5)$$\left[\mathrm{Mn}\left(\mathrm{III}\right)\mathrm{TCPP}\right]={\left[\mathrm{Mn}\left(\mathrm{II}\right)\right]}_{\mathrm{initial}}\left(1-e^{-k_{\mathrm{Mn}\left(\mathrm{II}\right)}t}\right)+{\left[\mathrm{Mn}\left(\mathrm{III}\right)\right]}_{\mathrm{initial}}\left(1-e^{-k_{\mathrm{Mn}\left(\mathrm{III}\right)}t}\right)$$

The different kinetics of these two different reactions were invoked to justify using this method to quantify simultaneously Mn(III)–TCPP generated from Mn(II) substitution and oxidation and Mn(III)–TCPP generated from Mn(III) exchange [29]. In this approach, the kinetic profiles are fit and deconvolved, considering them as the weighted sums of two independent exponentials [29]. However, this yields a classically ill-posed problem in applied mathematics for which accurate numerical schemes have been challenging to achieve—particularly with experimental data wherein small variations in the data can lead to substantial differences in the parameters achieved by fitting [30,31]. This problem becomes even more acute in samples containing more than one Mn-bearing species if the number of species, their reaction rates, or their concentrations are unknown a priori:(6)$$\begin{array}{ll} \;\left[\mathrm{Mn}\left(\mathrm{III}\right)\mathrm{TCPP}\right]= & {\left[\mathrm{Mn}\left({\mathrm{species}}_{1}\right)\right]}_{\mathrm{initial}}\left(1-e^{-k_{1}t}\right)\\ & +{\left[\mathrm{Mn}\left({\mathrm{species}}_{2}\right)\right]}_{\mathrm{initial}}\left(1-e^{-k_{2}t}\right)\\ & +{\left[\mathrm{Mn}\left({\mathrm{species}}_{3}\right)\right]}_{\mathrm{initial}}\left(1-e^{-k_{3}t}\right)+ \end{array}$$
(7)$$\left[\mathrm{Mn}\left(\mathrm{III}\right)\mathrm{TCPP}\right]=\overset{n}{\underset{1}{\sum }}{\left[\mathrm{Mn}({\mathrm{species}}_{n})\right]}_{\mathrm{initial}}\left(1-e^{-k_{n}t}\right)$$

Even under the assumption that the only Mn(III) chemistry occurring in these reactions is the proposed mechanism of a simple ligand-exchange reaction, deconvolving reactions from multiple unknown Mn complexes is a formidable challenge.

Using both the LBB and TCPP methods, differences in the responses from standard solutions of different Mn(III) complexes have been observed [24,32]. With the TCPP method, it was suggested that the reaction kinetics could be further resolved to distinguish the strength of ligand binding in different Mn(III) complexes based on their rate of exchange [32]. With the LBB method, a similar argument based on strong vs. weak ligand binding was suggested to explain the observation that some Mn(III) complexes react readily with LBB, while others do not react at all [24]. Building on these interpretations, DFOB has been described as a prototypical “strong ligand” and used for ligand exchange extractions to identify the fraction of soluble Mn(III) that is complexed to a weaker ligand [33].

This “strong ligand” paradigm likely has roots in the use of siderophore complexation reactions (including with DFOB) to determine the concentrations of Fe(III) in natural waters. In such cases, forward- and reverse-complexation reaction rates can be used to infer conditional equilibrium constants. This relationship has been empirically validated and works well for cases of simple single-step reversible substitution reactions, which Fe(III) complexations frequently are. However, with multistep complexation reactions, directly relating kinetics to thermodynamics is not possible. For example, the presence of calcium (Ca) slows down copper (Cu) chelation in seawater [34]; therefore, inferring the thermodynamics of Cu coordination based on reaction rate in the presence of Ca would not be appropriate. Ca does not affect the equilibrium constant for Cu coordination; it changes the rate by introducing additional steps to the reaction. The fundamental differences in reactivity between Fe(III) and Mn(III) allow such approximations to be accessible for the relatively inert Fe(III) but not for the comparatively labile Mn(III). Mn(III) complexation reactions are not the simple single-step reactions that their Fe(III) counterparts would participate in. Therefore, Mn(III)–L reaction kinetics cannot be used as a reliable indication of ligand-binding strength. More broadly, methods, assumptions, and interpretations that are based on the dynamics of Fe(III) should be reevaluated prior to their application to Mn(III).

Here we present the results of a suite of experiments designed to probe the utility of these approaches for measuring Mn(III) complexes in natural samples. Our results indicated important issues with each of these methods and interpretations of the data generated by using them. In the case of LBB, we observed that the assay—which requires a low-pH solution—is incompatible with Mn(III) complexes that degrade at low pH faster than they can react with LBB; this was illustrated in our experiments by Mn(III)–DFOB. In the case of the TCPP method, we found that the reactivity of Mn(III) species confounded the proposed reaction mechanism, such that fitting different kinetic profiles cannot reliably be used to quantify or draw robust conclusions about the nature of Mn speciation in unknown samples. In the case of ligand-exchange extractions, we caution that, since DFOB is not a redox stable ligand and its reactivity exhibits a strong pH dependence [17], this compound (and others like it) should not be used as reference species to examine thermodynamic binding strength of unknown molecules in environmental samples. In all cases, these issues stemmed from the inherent reactivity and lability of Mn(III)–L complexes.

## 2. Materials and Methods

### 2.1. Reagents

All chemicals were obtained from Sigma-Aldrich at the best available purity. All solutions were prepared with ultrapure water (UW) obtained from a Merck Millipore MQ Direct 8 water-purification system. Mn(II) chloride (MnCl_2_), potassium permanganate (KMnO_4_), and Mn(III)–acetylacetonate (acac) solutions were prepared by dissolution in UW. The Mn(III)–PP and Mn(III)–DFOB complexes were synthesized following the protocols from Madison et al. [29].

### 2.2. LBB Method

A 0.04% LBB solution was prepared in UW with 1% glacial acetic acid, equivalent to the primary reagent described in Jones et al. [24]. The pH of this solution was ~3. We report LBB concentration as a weight percent rather than molarity, as the low purity of commercially available LBB makes precise concentrations of dye content unreliable. Therefore, standard curves with KMnO_4_ must be employed with each batch of LBB reagent to calibrate quantitation for sample unknowns.

Reactions were performed in a 1 cm UV cuvette to monitor UV–vis spectral changes of reaction solutions. For the cuvette-based assay, similar to that described in Jones et al. [24], 40 µL of LBB solution was added to 2 mL of sample solution. For the reaction with Mn(III)–DFOB, we used a 6 µM Mn(III)–DFOB solution and monitored changes in absorbance at 310 nm for [Mn(III)–DFOB] and 624 nm for LBB oxidation. The final pH remained at ~3 for reactions with KMnO_4_ and increased slightly to ~3.5 with Mn(III)–DFOB. To assess the role of the low-pH reaction solution in Mn(III)–DFOB decomposition, a similar experiment was conducted with 2 µL of glacial acetic acid added to 2 mL of 6 µM Mn(III)–DFOB (final pH ~ 3.3).

For a higher throughput (albeit lower sensitivity) LBB assay, reactions can be performed in a 96-well plate. For the plate-based assay, we follow a protocol similar to that described in numerous previous studies [20,21,22]. For samples including biomass, sediment, or other particulate matter, we reacted 250 µL of LBB solution with 50 µL of sample in a microcentrifuge tube for 15 min to allow the reaction to proceed to completion, centrifuged to remove particulate matter, and then transferred 250 µL of supernatant to a plate for quantification. In this study, the plate-based assay was only used with a KMnO_4_ standard solution and therefore did not require a centrifugation step, so 208.3 µL of LBB solution and 41.6 µL of sample were reacted directly in the plate.

To determine the pH threshold of LBB activity, the LBB reagent solution was diluted 10× and adjusted to pH values ranging from 3 to 8.2, and then reacted with 50 µM KMnO_4_ in the plate-based assay.

Cuvette-based UV–vis spectra were recorded on a Hewlett Packard 8454 diode array spectrophotometer. Ninety-six-well plates were read with a BioTek Cytation 5 plate reader.

### 2.3. TCPP Method

The TCPP method was performed by using the protocol from Madison et al. [29]. A 0.2 mM TCPP solution was prepared in UW with 0.1 M NaOH. To guard against potential photochemistry, the bottle was wrapped in aluminum foil. A 12 mM CdCl_2_ solution was prepared in UW. The buffer solution was prepared with 0.025 M sodium tetraborate, 0.1 M HCl, and 0.6 M imidazole; the pH was adjusted to 8.0 with 3 M HCl. Reactions were performed in a 1 cm UV cuvette. The precursor, Cd(II)–TCPP, was prepared by addition of 360 μL of TCPP solution, 6 μL of CdCl_2_ solution, and 120 μL of buffer solution; and then it was brought up to 3 mL with UW to yield 24 μM Cd(II)–TCPP. A total of 10 μL of 0.54 mM Mn solution was added into the precursor solution for the reaction, for a final concentration of 1.8 μM Mn solution. Changes in absorbance were monitored at 468 nm for [Mn(III)–TCPP]. Reactions were run in triplicate; the data reported are a representative example. For reactions under argon, each solution was purged with stirring for 15 min, which scrubs most but not all dissolved O_2_ from solution, and reactions were conducted under an argon atmosphere in a two neck UV cell.

## 3. Results and Discussion

### 3.1. Limitations of the LBB Method

It has been reported that some Mn(III)–L complexes, most notably Mn(III)–DFOB, do not react with LBB [24]. This observation was interpreted in thermodynamic terms as a reflection of the binding strength of Mn(III)–L complexes—that weakly bound complexes react readily, while strongly bound complexes react slowly or not at all [24,29,32]. However, LBB reacts readily with a wide range of solid Mn oxide phases [23], which are much more stable than any Mn(III)–L complex. Therefore, we investigated an alternative explanation for the reported data.

The LBB method is typically carried out in an acetic acid solution, at pH ~3 [20,21,22,24]. However, the Mn(III)–DFOB complex is known to persist only in solution within the pH range of ~7–11 [17]. We attempted to measure Mn(III)–DFOB by using LBB and did not observe the spectral change at 624 nm characteristic of LBB oxidation (Figure 2A), as is consistent with previous reports that Mn(III)–DFOB does not react with LBB [24]. However, by also monitoring the absorbance spectrum at 310 nm for the characteristic absorbance of Mn(III)–DFOB, we found that the Mn(III)–DFOB disappeared on a timescale of seconds upon introduction to the reaction solution of the LBB assay. The same phenomenon occurred when acetic acid without LBB was introduced to the Mn(III)–DFOB solution (Figure 2B). These results demonstrated that the lack of reaction with LBB occurred not because the Mn was so strongly bound that the LBB could not access it, but rather because the complex disintegrated in the low-pH reaction solution before it had an opportunity to react with LBB. Since the mechanism of decomposition reduces Mn(III) to Mn(II) [17], which does not react with LBB, this explained why the LBB assay does not detect Mn(III)–DFOB.

We further studied the pH dependence of the LBB method by using standard solutions of KMnO_4_ and showed that, above pH 5, there is substantial loss of LBB signal (Figure 2C). Therefore, we concluded that, since a low pH reaction solution is a methodological requirement, the LBB method is not suitable for quantitation of high-valent Mn species that decompose rapidly at a low pH. This may include not only Mn(III)–DFOB, but a variety of other Mn(III)–L complexes that could exist in the environment. As an example, at the mean pH of Black Sea surface waters (8.38) [35], Mn(III)–DFOB would be stable for weeks. Since complexation to DFOB and other siderophores is thought to increase Mn bioavailability [36], such complexes are likely important players in Mn biogeochemistry. However, any such complexes are undetectable by the LBB method if, similar to Mn(III)–DFOB, they decompose in the assay faster than the assay can report their presence.

In spite of this limitation, the LBB method remains a very valuable assay, particularly in contexts such as the study of biological Mn oxidation processes [20,21,22] or the detection of Mn oxides in unknown materials [23,33,37]. When using LBB to detect soluble species, it must be understood that, since LBB can only access a subset of Mn(III)–L complexes, an additional LBB-invisible pool of Mn(III) may exist. This is not a reflection of ligand-binding strength; rather, it arises from redox-driven chemical reactions occurring during the assay. Because of this limitation, data quantifying soluble Mn(III) species via the LBB method likely systematically underestimate the true abundance of these species in natural samples.

### 3.2. Behavior of the TCPP Method

The use of the TCPP method to measure and distinguish between Mn species was originally shown with standard solutions of MnCl_2_ as a source of hexaaqua–Mn(II), Mn(III)–PP as a fast-reacting Mn(III) complex, and Mn(III)–DFOB as a slow-reacting Mn(III) complex [29]. In the case of hexaaqua–Mn(II), the Mn(III)–TCPP absorption signal fully developed within 1 min; with the Mn(III) complexes, the signal from Mn(III)–PP developed over several minutes, while the signal from Mn(III)–DFOB developed even more slowly, not reaching the absorbance maximum during the 15 min time course examined. We reproduced these phenomena and additionally examined the Mn(III)–acac complex, which showed kinetic behavior similar to that of Mn(III)–PP (Figure 1B and Figure 3A).

The reactive nature of Mn(III)–L complexes, in particular, the known redox instability of Mn(III) complexed to organic ligands, such as DFOB [17], caused us to reconsider the proposed mechanism of this method for detecting Mn(III). If the reaction between Mn(III) complexes and TCPP is well described as a simple ligand exchange, not involving any redox chemistry, it should not be affected by the availability of O_2_. Therefore, we performed the same reactions under argon rather than in air. The reaction with Mn(III)–PP behaved consistently with the proposed mechanism, showing no meaningful difference under argon. However, with both Mn(III)–acac and Mn(III)–DFOB, the reaction behaved strangely under argon (Figure 3B). The absorbance fluctuated irregularly with time, and with Mn(III)–DFOB—which, in air, did not reach its stoichiometric absorbance maximum during the time course examined—the reaction appeared to proceed at a much higher rate, surpassing the maximum expected absorbance in minutes. These observations appeared inconsistent with the idea that the chemistry taking place in these reactions is just a simple substitution reaction.

The use of the TCPP method as a quantitative assay for soluble Mn(III) depends on understanding the reactions taking place in the solution. If additional reactions are occurring here, either instead of or in addition to the proposed mechanism, then caution should be taken when interpreting results based on either the magnitude or rate of Mn(III)–TCPP production. The absorbance changes we observed in the Mn(III)–acac and Mn(III)–DFOB reactions suggested changes in the coordination environment of the Mn in the porphyrin, possibly attributable to ternary complex formation or coordination of exogenous ligands or oxidized ligand fragments. Whatever the mechanism, the chemical complexity implied by these observations undermines this method as an assay for environmental Mn(III). Our data suggested that this method may only be appropriate for Mn(III) complexes with redox stable ligands, such as Mn(III)–PP. Since the identity and distribution of ligands for soluble Mn(III) in natural samples remain largely unknown, this method may not accurately characterize such samples.

Even in the absence of sample unknown materials and with a well-behaved Mn(III)–L complex, interactions between Mn(III) and other reagents may complicate the solution chemistry and thereby confound this method. For example, imidazole is used in the method as a buffer to facilitate metal complex substitution. It has been shown that varying the concentration of the imidazole changes the reaction kinetics [38]. The fact that the kinetics can be modulated by an additional species demonstrates that the reaction between Mn(III)–L and TCPP cannot be a simple single-step substitution, and, therefore, inferring thermodynamic properties from kinetic behavior is not appropriate. With both Mn(III)–PP and Mn(III)–acac, we observed absorption changes upon the introduction of the Mn species to the imidazole buffer without TCPP, indicating that these Mn(III) complexes reacted in the presence of this reagent. Interactions between Mn materials and additional species—including but not limited to the imidazole buffer—confound interpretation of the data generated by this measurement. With known materials, one can potentially constrain the suite of possible reactions occurring during the assay; with natural samples of unknown composition, however, constraining all possible reactions and their products becomes a much bigger challenge.

### 3.3. Ligand-Exchange Extractions Using DFOB

Based on its slow exchange in the TCPP method and lack of reaction with LBB, Mn(III)–DFOB has received attention as a prototypical “strongly binding, non-reactive” Mn(III) complex. However, this interpretive framework does not differentiate between kinetics and thermodynamics—the binding strength is a thermodynamic issue, and the rate of exchange or reaction is a kinetic issue. While the Mn(III)–DFOB complex does have a high stability constant, reactivity or perceived lack thereof from unknown complexes in the TCPP or LBB methods does not necessarily indicate ligand-binding strength; and known binding strengths do not necessarily predict reaction rates or mechanisms.

As the prototypical strong ligand, DFOB has been used for ligand-exchange extractions of soluble Mn(III)–L complexes from natural samples [33]. However, we caution that ligand-exchange extractions with a ligand that is not redox stable—such as DFOB [17]—may produce misleading results. Furthermore, the products of Mn(III)–DFOB degradation can include chromophores with similar or overlapping absorption features to Mn(III)–DFOB itself [17]; these can additionally confound the measurement.

## 4. Conclusions

Results from these methods for detecting and characterizing soluble Mn(III)–L complexes—the TCPP method, in particular—have underpinned the field of manganese aquatic chemistry for the last decade. Although our data raise concerns about the validity of this method, we stress that the work using it to demonstrate the widespread presence of soluble Mn(III) was groundbreaking and instrumental in shifting the field away from the previous paradigm that environmental manganese speciation was a simple dichotomy of soluble Mn(II) and insoluble Mn oxides [3,4,5]. Now that we recognize the potential significance of soluble Mn(III) complexes as reactive intermediates in critical biogeochemical processes, we as a community have our work cut out for us to understand better their diversity, fluxes, and precise roles in environmental chemistry. In pursuing this better understanding, we should appreciate the highly reactive nature of these complexes that, at once, makes them fascinating, important, and so difficult to study. We need to appreciate the limitations of what our current tools can and cannot constrain in order to apply them most effectively, while working to develop better approaches.

## Figures and Tables

**Figure 1 molecules-27-01661-f001:**
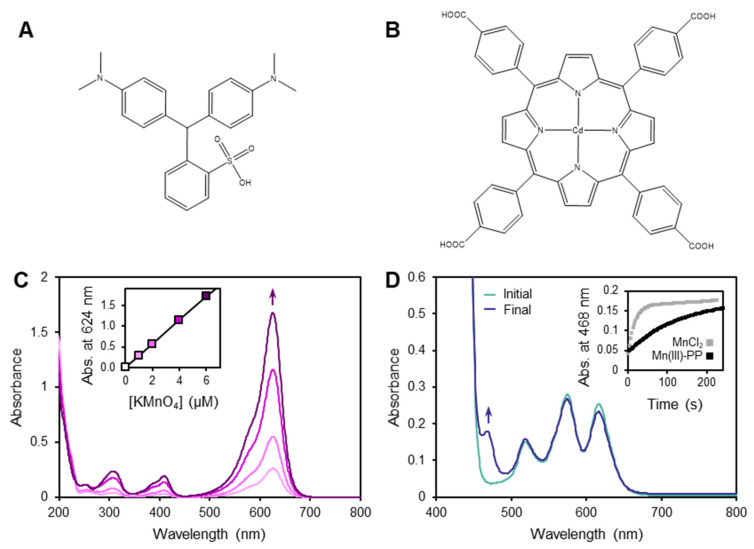
Leucoberbelin blue (LBB) and Cd-porphyrin (TCPP) spectrophotometric methods for detecting manganese species. (**A**,**B**) Chemical structures of the LBB (**A**) and Cd(II)–TCPP (**B**) reagents. (**C**,**D**) UV–vis absorbance spectra illustrating the application of these methods. (**C**) LBB method on a standard curve of KMnO4 solutions, with the inset showing the linear absorbance change at 624 nm with KMnO_4_ concentration due to oxidation of LBB. (**D**) TCPP method on a standard solution of MnCl_2_, showing the change in absorbance at 468 nm from Mn(II) substitution and oxidation to generate Mn(III)–TCPP. Inset shows the kinetic profile of this reaction, along with the Mn(III)–PP ligand-exchange reaction to generate Mn(III)–TCPP.

**Figure 2 molecules-27-01661-f002:**
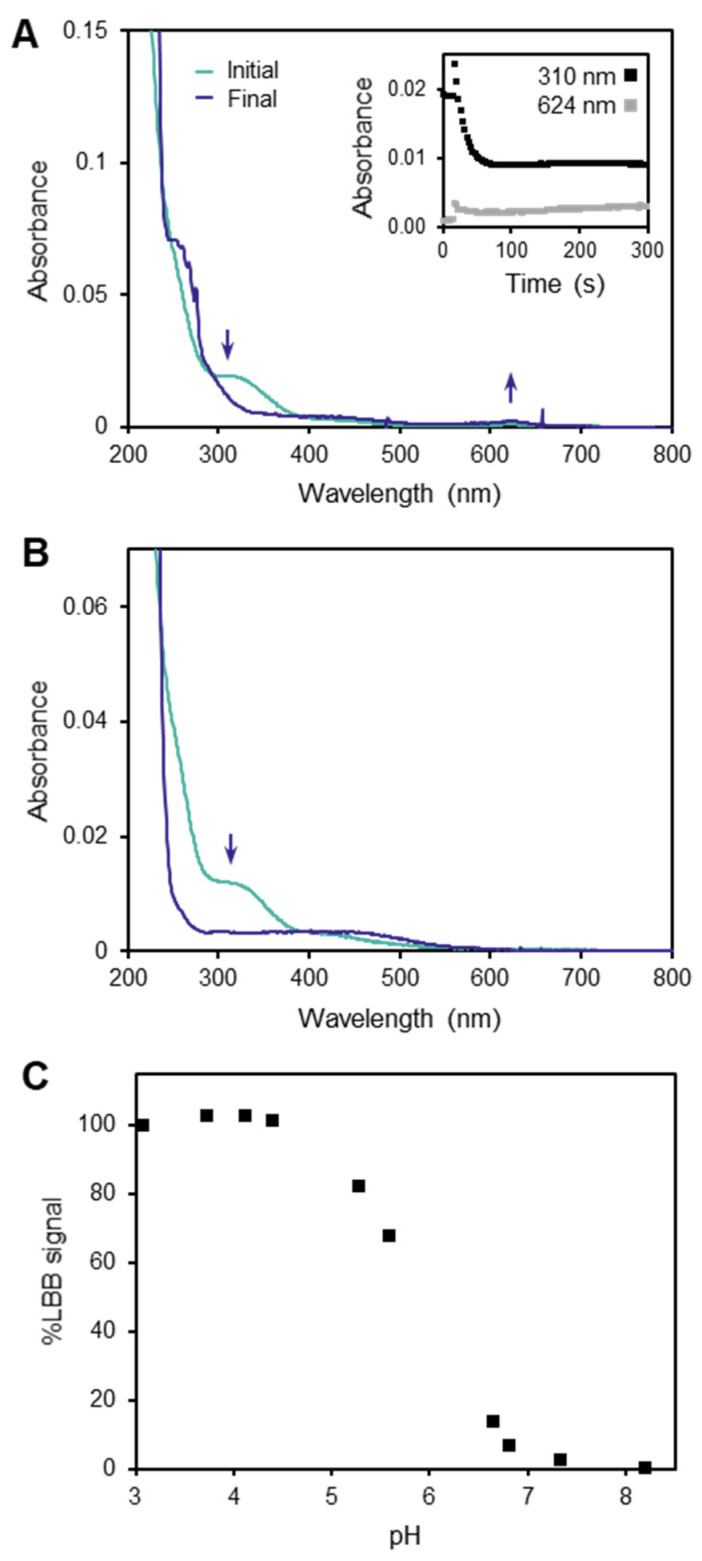
Mn(III)–DFOB is incompatible with the LBB method, due to its rapid decomposition at low pH, a methodological requirement for the LBB assay. (**A**) LBB solution added to 6 µM Mn(III)–DFOB solution. The characteristic Mn(III)–DFOB band at 310 nm disappeared on a timescale of seconds, and the oxidized LBB band at 624 nm did not appear. (**B**) One percent acetic acid without LBB added to 6 µM Mn(III)–DFOB solution. The Mn(III)–DFOB band still disappeared, indicating that the Mn(III)–DFOB decomposition was caused by the change in pH rather than any reaction with LBB. (**C**) LBB signal with KMnO_4_ drops off in reaction solutions above pH 5, demonstrating that a low pH reaction solution is required for this method. Percent LBB signal reports the absorbance at 624 nm relative to the reaction at pH 3, the baseline solution pH for this method.

**Figure 3 molecules-27-01661-f003:**
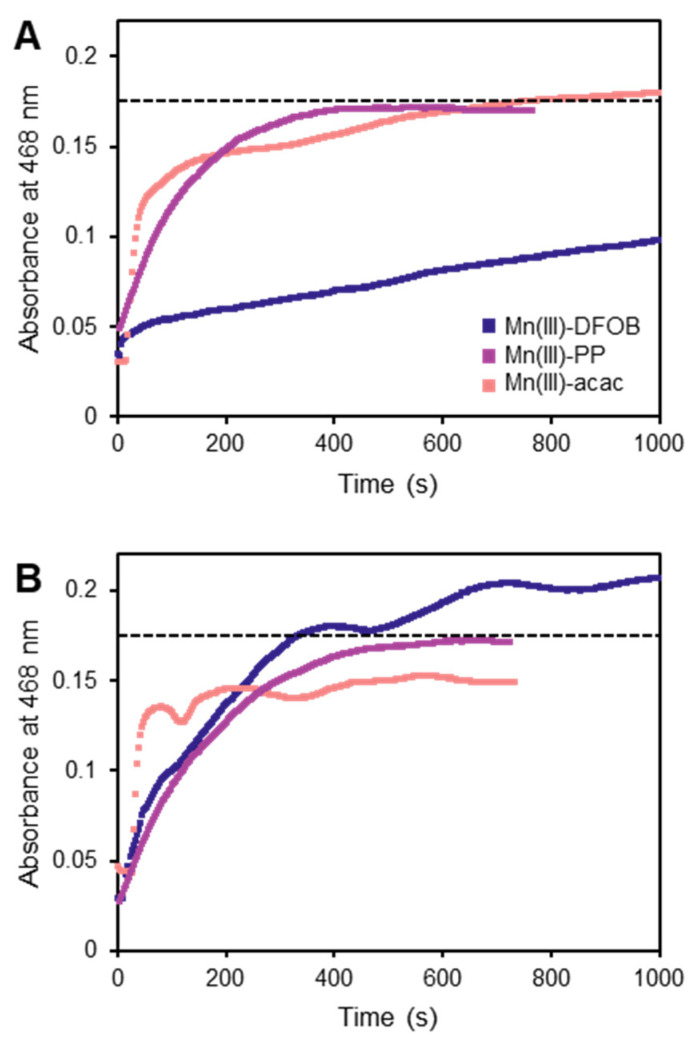
Reactions of Cd(II)–TCPP with Mn(III) complexes, both under air (**A**) and under argon (**B**), raise concerns about the proposed mechanism of this method. Dashed lines indicate maximum absorbance expected from 1.8 µM Mn(III)–TCPP. (**A**) Under air, all three reactions display absorbance increasing with time monotonically. Mn(III)–DFOB reacts much more slowly than Mn(III)–PP, as previously reported. (**B**) Under argon, the Mn(III)–PP reaction behaves the same as under air. However, the Mn(III)–acac and Mn(III)–DFOB reactions display very different kinetic profiles, casting doubt on the proposed mechanism. With Mn(III)–acac, the increase in absorbance is no longer monotonic. With Mn(III)–DFOB, the reaction proceeds far more rapidly and exceeds the maximum expected absorbance.

## Data Availability

Data are reported in the figures. Raw data available upon request.

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
