# Peer review of "Challenges of Measuring Soluble Mn(III) Species in Natural Samples"

_molecules, 2022, doi:10.3390/molecules27051661_

Round 1

Reviewer 1 Report

Review of “Challenges of Measuring Soluble Mn(III) Species in Natural Samples”.  Kim et al. presented experimental results that may demonstrate limits of common analytical methods used to quantify soluble Mn(III)-L complexes. They have many interesting findings as claimed in Abstract and I’m not going to repeat here. The manuscript is generally well written, and related scientific questions are important. I commend authors for such nice work. I only have a few comments that may be helpful for authors when they revise the manuscript. My major concern is that the Introduction is too long to focus the important information of the manuscript. I don’t think that make the introduction be a literature review is a proper way. I highly recommend authors significantly reduce the length of Introduction and focus the scientific questions of your study. In particular, the calculations should not occur in Introduction. Another concern is that the title of the manuscript— seems that the manuscript is a literature review while the truth is not. I suggest authors revise it to make it better reflect the content of the manuscript.

Author Response

We would like to thank the reviewer for their kind words and helpful feedback. Based on this feedback, we attempted to rearrange the content. We tried removing some material and moving other material such as the calculations to the results and discussion sections. However, we felt that these changes made the manuscript less useful for its intended purpose, which is to take stock of the current state of knowledge concerning Mn(III) complexes in nature and methods for detecting them, in order to highlight the paradigm shift and further method development required for future investigations into these chemical species and their critical roles in biogeochemistry. Thus, while we agree that this paper blurs the lines between a primary research article and a literature review, we argue that that is part of what makes it useful, and furthermore, what makes it particularly appropriate for this special issue on manganese.

Reviewer 2 Report

Pleased see attached file

Author Response

The reviewer felt that we were too critical (specifically lines 137-140), suggesting that we simply wished to devalue these methods. This was not our intention; therefore, we have changed the wording of that section to tone it down. Detailed please see attached file.

Reviewer 3 Report

The ms by Kim et al. is clearly and well written. It gives a very valuable insight into the complicated world of Mn-speciation, its redox chemistry and especially Mn(III) complexes and actually in this case the title is very fitting although it doesn’t give away the conclusions yet. It is an interesting inorganic view into today’s most commonly used Mn(III) method- the leucoberbelin blue method and its limitations – decomposition under low pH. In my opinion it is a very good paper as it directly addresses issues with this Mn(III) method that were troubling but not yet enough tested even though the method has been commonly used. This might have led to an overestimation of Mn(III) ligands over inorganic Mn(II) in some locations.

Overall, in some detail,I think it sometimes could even go into more detailed redox and coordination chemistry (crystal field theory, orbitals) for these Mn-complexes as this is rare in marine trace metal chemistry and it use more reaction equations to facilitate the understanding and reproducibility for the reader by giving exact pH and amounts of reagents. The methodological details could be improved along with some of the figures (lines, their thickness and errors) as well.

Minor comments below:

L15 introduce Manganese along with other chemical species, please check the ms for this and its uniform use.

L55 use the ferric iron to introduce the species Fe(III).

L112 O2 2 needs to be lower case and oxygen as O2 hasn’t been introduced yet although it and ROS are very important in the redox cycle and reactions of Mn.

L159 Ca should be spelled out at the beginning of the sentence and introduced, some for Cu. By mentioning Ca as a lot of marine articles are cited, the importance of Ca in that environment and also the buffer capacity of seawater could be mentioned and if this has been performed in saline waters- what are the effects of salinity and other trace metals interfering in this reaction?

L177-180 Reference(s) needed to justify these statements

L186 MnCl2 should be spelled out at the beginning of the sentence and introduced and lower case 2

L194 KMnO4 lower case 4 as well, please, and not introduced yet

Section 2.2. L189 onwards: what about error, standard deviations, accuracy and especially the LOD of the method? This is missing completely in the ms, but necessary.

L193 KMnO4 lower case 4 as well, please

L232 same comment as in L112

L262-264 please give exact pH of the reaction and/or the concentration and amount of acetic acid added

Figure 2C please add errors into the figure

L298 Figure 3 not great looking figures: very thick lines and colours that are not well visible for colourblind, please change the graphs, please – why is the x-axis in seconds if the text is in minutes, then please change the x-axis to minutes as well as this is better visible and understandable for the reader. Also use the same colours in figure 1b for the same species shown here, please and reduced the line thickness in figure 1b as well.

L301 “Under air” in contrast to under argon- in my opinion the whole aspect of the influence of oxygen is missing in this ms and with it some redox reaction equations to demonstrate the differences and the electron transfer happening in the reaction – this is missing in L310 for example as well to demonstrate the reader to reconsider the proposed mechanism along with the authors without having to look up all of the reactions and read all the background papers. This counts for the next sentence as well, please show the reaction with oxygen and why it the mechanism should or shouldn’t be affected by it.

Author Response

We would like to thank the reviewer for their kind words and detailed feedback. We addressed their specific minor comments in the text, and thank them for catching those points. We also modified the figures based on their suggestions, specifically making the markers smaller to address their line thickness issues. However, we did not change the colors, because the colors we chose were based on a colorblind friendly palate (the Tol palate from https://davidmathlogic.com/colorblind). The reviewer made good suggestions concerning crystal field theory and orbitals that we would love to address in a future publication, but we believe is outside the scope of the current manuscript. Regarding methodological details, we have given pH values to the precision with which we measured them when we did this work (using pH strips). Given that our methods section is intended to provide the details needed to reproduce the work that we did (this is not the proposal of a new analytical procedure), we believe that we have provided sufficient details. We also appreciate the reviewer’s frustration that we are unable to provide a satisfying explanation for the effect of oxygen on the TCPP method (their comment concerning line 301). We too felt similarly frustrated by this result. However, we chose not to go down the rabbit hole of figuring out exactly what mechanism was going on, given that the point here was that additional chemistry beyond the originally proposed mechanism is occurring, which confounds the use of this method to measure Mn(III) species.